Bite and tooth marks on sauropod dinosaurs from the Morrison Formation

Lei Roberto 1
Tschopp Emanuel 2 3 4
Hendrickx Christophe 5
Wedel Mathew J. 6
Norell Mark 2
Hone David W.E. dwe_hone@yahoo.com 7
1 Università degli Studi di Modena e Reggio Emilia , Moderna , Italy
2 American Museum of Natural History , New York , NY , USA
3 Universität Hamburg , Hamburg , Germany
4 GeoBioTec, Faculdade de Ciencias e Tecnologia, Universidade Nova de Lisboa , Caparica , Portugal
5 Unidad Ejecutora Lillo, CONICET-Fundación Miguel Lillo , Tucumán , Argentina
6 Western University of Health Sciences , Pomona , CA , USA
7 School of Biological and Behavioural Sciences, Queen Mary University of London , London , United Kingdom
Young Mark
Electronic publication date: 2023 Nov 14
Publication date: 2023
Volume: 11
Electronic Location ID: e16327
Received 2023 May 1; Accepted 2023 Sep 30
Copyright: ©2023 Lei et al.
Copyright year: 2023
Copyright holder: Lei et al.
License: This is an open access article distributed under the terms of the Creative Commons Attribution License, which permits unrestricted use, distribution, reproduction and adaptation in any medium and for any purpose provided that it is properly attributed. For attribution, the original author(s), title, publication source (PeerJ) and either DOI or URL of the article must be cited.
License URL: https://creativecommons.org/licenses/by/4.0/

Keywords: Sauropoda, Theropoda, Dinosauria, Predator-Prey, Ichnology

Funding: The Consejo Nacional de Investigaciones Científicas y Técnicas (CONICET) Agencia Nacional de Promoción Científica y Tecnológica, Argentina to Christophe Hendrickx (Beca Pos-doctoral CONICET Legajo ) 181417 Emanuel Tschopp held a Theodore Roosevelt Memorial Fund and Division of Paleontology Postdoctoral Fellowship provided by Richard Gilder Graduate School at the American Museum of Natural History, New York Christophe Hendrickx was funded by the Consejo Nacional de Investigaciones Científicas y Técnicas (CONICET) and Agencia Nacional de Promoción Científica y Tecnológica, Argentina to Christophe Hendrickx (Beca Pos-doctoral CONICET Legajo 181417). Emanuel Tschopp held a Theodore Roosevelt Memorial Fund and Division of Paleontology Postdoctoral Fellowship provided by Richard Gilder Graduate School at the American Museum of Natural History, New York, during the course of this work at AMNH, and currently holds a Alexander von Humboldt Fellowship for Experienced Researchers. The funders had no role in study design, data collection and analysis, decision to publish, or preparation of the manuscript.

==============================
Tooth-marked bones provide important evidence for feeding choices made by extinct carnivorous animals. In the case of the dinosaurs, most bite traces are attributed to the large and robust osteophagous tyrannosaurs, but those of other large carnivores remain underreported. Here we report on an extensive survey of the literature and some fossil collections cataloging a large number of sauropod bones (68) from the Upper Jurassic Morrison Formation of the USA that bear bite traces that can be attributed to theropods. We find that such bites on large sauropods, although less common than in tyrannosaur-dominated faunas, are known in large numbers from the Morrison Formation, and that none of the observed traces showed evidence of healing. The presence of tooth wear in non-tyrannosaur theropods further shows that they were biting into bone, but it remains difficult to assign individual bite traces to theropod taxa in the presence of multiple credible candidate biters. The widespread occurrence of bite traces without evidence of perimortem bites or healed bite traces, and of theropod tooth wear in Morrison Formation taxa suggests preferential feeding by theropods on juvenile sauropods, and likely scavenging of large-sized sauropod carcasses.

Introduction

Tooth traces are a form of trace fossil produced by contact between a tooth and a bone, typically during feeding. They yield data regarding the biology and behavior of the bite-making animal and potentially about interactions between extinct species. Moreover, the analysis of multiple tooth traces may provide information about the ecological relationships existing between animals, such as non-avian dinosaurs (e.g., Fiorillo, 1991; Jacobsen, 1998; Hone & Chure, 2018).

Despite their potential importance, tooth traces from carnivorous theropods are generally poorly studied, with a handful of specimens described in detail and only two systematic surveys performed to date (see Jacobsen, 1998; Drumheller et al., 2020). This could be partly due to the rarity of such traces, which suggests that contact between bones and theropod teeth was often accidental (see Hone & Rauhut, 2010), and it was also hypothesized that the dentition of most taxa was not suited for processing such a hard tissue (Fiorillo, 1991). The only notable exception is the tyrannosaurids; numerous bite traces are known from formations where these are the dominant carnivorous clade (e.g., Fiorillo, 1991; Jacobsen, 1998; Hone & Rauhut, 2010). Of the few detailed descriptions of theropod tooth traces, many are those made by tyrannosaurs (e.g., Erickson & Olson, 1996; Hone & Watabe, 2010; De Palma et al., 2013), which leave traces more often than other theropods (Hone & Rauhut, 2010) such that these are perhaps the best understood clade in terms of their feeding patterns and ecology.

Tooth traces take different shapes depending on the behavior of the biting animal. For instance, elongated traces are produced by tooth dragging, whereas the collapse of the bone surface indicates particularly strong bites (D’Amore & Blumenschine, 2009). In some cases, it is also possible to infer information about the carnivore’s size, which may be predicted from parallel traces, left by a single bite. Here, the distance separating them should reflect the spacing of the teeth, although several factors may bias the results and render it difficult to correctly assign a trace maker (see Hone & Chure, 2018; Brown, Tanke & Hone, 2021).

Further information that may be inferred from tooth traces includes the ecological relationships between herbivores and carnivores. For example, prey preferences could be reflected by a majority of feeding traces associated with a particular taxon (Jacobsen, 1998) or potentially from the proportions of taxa with healed bites. However, it remains mostly impossible to distinguish between predation or scavenging events, hampering interpretations of interactions between living members of prey and predator species (e.g., see Currie & Jacobsen, 1995; Holtz, 2003; Bader, Hasiotis & Martin, 2009; Hone, Tanke & Brown, 2018). In some cases, failed hunting attempts were identified thanks to the presence of healing tissues in correspondence to the bites (De Palma et al., 2013). Otherwise, the location of the traces may favor one or the other interpretation (Hunt et al., 1994). One of the most interesting ecosystems to understand such carnivore-consumed (sensu Hone & Tanke, 2015) relationships is the terrestrial Upper Jurassic Morrison Formation in the western United States (Drumheller et al., 2020).

The Morrison Formation is an extremely large unit in the central to western part of North America (Turner & Peterson, 2004). It is mainly composed of continental sediments forming an alluvial plain, which is thought to have been covered in rivers and lakes (Foster, 2003; Turner & Peterson, 2004). For at least the last four decades, the reigning paradigm of the paleoenvironment of the Morrison Formation has been a seasonally dry to semi-arid floodplain, traversed by numerous river channels and dotted with ponds and lakes; some of the latter were ephemeral, alkaline, or saline (see Dodson et al., 1980, Engelmann, Chure & Fiorillo, 2004; Turner & Peterson, 2004; Richmond, Hunt & Cifelli, 2020, and references therein). In this scenario, forests were constrained to riparian corridors that crossed the essentially savannah-like floodplains (Demko & Parrish, 1998; Parrish, Peterson & Turner, 2004). In contrast, fluvial and lacustrine facies, aquatic vertebrate and invertebrate fossils, and paleo wood have been considered by some authors to indicate wetter and more equable conditions in the Morrison depositional basin (Tidwell, 1990; Gee et al., 2019; see Richmond et al., 2019 for a brief review), with drought-induced die-offs of dinosaurs and other terrestrial and freshwater organisms (Richmond & Morris, 1998; Carpenter, 2013; Richmond, Hunt & Cifelli, 2020) as occasional punctuation marks. However, given the long period of deposition of approximately 10 My, and the vastness of the depositional basin, which covers about 1.2 million km2 (Maidment & Muxworthy, 2019), it is also possible that both conditions occurred, and that the climate varied in time and space from rather wet to seasonally dry and arid conditions on a cyclical basis (Tschopp et al., 2020).

Whether the Morrison paleoenvironment was basically dry but well-supplied by rivers and lakes, or wetter but stricken by periodic droughts, it was apparently a favorable environment for sauropod dinosaurs. Numerous Morrison localities have produced remains of three or four coexisting sauropod genera (Foster, 2021), and the Dry Mesa Dinosaur Quarry in western Colorado has at least six distinct genera of sauropods represented (Curtice et al., 2023). Potential resource partitioning among Morrison sauropods is indicated by many aspects of neck, skull, and tooth morphology, as well as tooth wear and tooth replacement rate (Fiorillo, 1998; Whitlock, 2011; Button, Rayfield & Barrett, 2014; McHugh, 2018). Both tooth wear and tooth morphology provide evidence of ontogenetic shifts in diet (Fiorillo, 1998; Woodruff et al., 2018), which may have been especially important for animals that passed through five orders of magnitude during their ontogeny (Sander et al., 2011).

Large sauropods such as Camarasaurus, Diplodocus, Apatosaurus, and Brachiosaurus in fact dominated the Morrison ecosystem in terms of abundance and body size (Chure et al., 2006; Farlow, Coroian & Foster, 2010; Whitlock, Trujillo & Hanik, 2018; Foster, 2020; Mannion, Tschopp & Whitlock, 2021). There are also numerous large-bodied (>5m in body length) theropods including Allosaurus, Ceratosaurus, and Torvosaurus (Gilmore, 1920; Henderson, 1998; Bakker & Bir, 2004; Foster, 2020) which would have fed upon them, whether through hunting them as prey (as adult or juveniles) or scavenging from carcasses.

Several tooth-marked bones belonging to sauropods from the Morrison Formation have been already described in detail (e.g., Hunt et al., 1994; Chure, Fiorillo & Jacobsen, 1998; Hone & Chure, 2018; Drumheller et al., 2020) but this remains an area of limited study given the huge numbers of bones that have been collected from this region. Such specimens represent a precious source of information to determine what may have eaten animals of this size given the discrepancy between most adult sauropods and contemporaneous theropods. Here we assess the interactions between theropods and sauropods in the Morrison Formation based on an examination of a total of 68 tooth-marked elements from 40 individual sauropods. We also find that wear on teeth of theropods was likely produced by feeding on sauropods and that tooth-on-bone contact was more frequent than previously thought. However, bites on large sauropods likely represent scavenging in most cases and predation and consumption of juvenile sauropods was probably more common.

Materials and Methods

We surveyed sauropod specimens from the Morrison Formation looking for possible tooth traces. Data was acquired both from the literature and visits to collections by several authors. Whereas several of the noted occurrences derive from rather incidental observations during collection visits for reasons other than a survey of bite marks, we were able to systematically go through more than 600 single bones of Morrison Formation sauropods at the American Museum of Natural History (AMNH; an approximated 80% of the cataloged collection, but note that several dozen to a few hundred bones from Howe Quarry and other localities remain unprepared and have not yet received a catalog number; see Tschopp, Mehling & Norell, 2020) with the aim of documenting any potential bite trace. Because of time constraints during the survey, and the sheer size of the collections at AMNH, the specimens were only analyzed with the naked eye, rotating the bones (where possible) to observe them under favorable lighting conditions. Hence, it is possible that smaller, less visible, traces, along with features like striations, which may have been observed with magnifying lenses (Blumenschine, Marean & Capaldo, 1996), remain underreported herein.

Surveyed sauropod specimens, be it from our active survey at the AMNH, reports from literature, or incidental observations during collection visits, cover the entire geographic and stratigraphic range of the Morrison Formation. Elements with recognized tooth traces are from 19 localities from Colorado, Montana, South Dakota, Utah, and Wyoming, mostly from the upper half of the formation, and cover a range of lithologies (see Data S1).

Two different classification methods, the first from Binford (1981) and D’Amore & Blumenschine (2009) and the second from Hone & Watabe (2010), were followed to describe tooth traces (see Data S2). Both approaches refer to four broad categories which mostly overlap. However, the first system aims at describing the trace’s morphology, while the second one focuses on the behavior that led to the bite. We decided to primarily use the following terms defined by Hone & Watabe (2010):

1. Drag (caused by tooth dragging, intact cortex); some traces were classified as drags despite collapse of the bone cortex because they were interpreted as shallow traces left on already damaged tissue.

2. Bite and drag (caused by tooth dragging, damaged cortex).

3. Pit (no dragging, intact cortex).

4. Puncture (no dragging, damaged cortex).

5. To this we add a fifth category - ‘Removed’ - to indicate the complete removal of a portion of a bone. Note that this is effectively a subdivision of the Bite and Drag and/or Puncture categories since the teeth have penetrated the cortex in order to remove the piece of bone (Fig. 1).

Bones with unambiguously identified bite traces were documented directly by the authors through either 3D scanning or photographs. Surface scans were acquired by means of an Artec 3D Spider scanner, whose micrometer precision allowed us to obtain high resolution digital reconstructions (see https://www.morphosource.org/projects/000518771).

Figure 1 Examples of bite traces on Morrison Formation sauropod elements.

Examples of each tooth trace category used for this article. (A) Drag on pubis of AMNH FARB 675 (currently considered an indeterminate macronarian 3D model of the holotype of “Apatosaurus” minimus, ©American Museum of Natural History); (B) bite and drag on a phalanx of AMNH FARB 264 (3D model; an indeterminate neosauropod, ©American Museum of Natural History); (C) pit on a mid-caudal vertebra of AMNH FARB 5760 (part of the topotype material of Camarasaurus supremus); (D) puncture on a metapodial of AMNH FARB 30116 (3D model; an indeterminate sauropod ©American Museum of Natural History); (E) removed on a fibula of AMNH FARB 582 (3D model; a Camarasaurus ©American Museum of Natural History).

Due to the excessive size or irregular shape, several specimens had to be photographed instead; for some elements the scanning software was not able to produce a satisfactory reconstruction, whereas other elements were too large, heavy, or fragile so moving and/or flipping them could have caused damage.

We determined the general shape of each trace, distinguishing between straight and curved traces.

For many specimens, we also provide measurements of length, width and depth of the bite traces. Moreover, in case of parallel traces (possibly left by the same bite), we also measured the spacing between one trace and the next one, along with the total length and width of the bite area (see Data S3 and Fig. 2). Whenever possible, traces were measured with calipers on the original specimens. Where appropriate, measurements were taken from photographs. The photographed traces were measured through the morphometric software tpsDig. To do so, we created a tps file from the picture with tpsUtil. Then, we placed landmarks on the specimen and measured the distance separating them with the meter tool. Being a supra-specific analysis, the error associated with 2D data from 3D objects was considered acceptable (Cardini, 2014; Courtenay et al., 2018). The 3D-scanned specimens were measured with Meshlab. Landmarks were placed in correspondence to bite traces; the distance between them was calculated by means of the Pythagoras theorem = SQRT (POWER (X2 −X1;2)+POWER (Y2 + Y1;2)+POWER (Z2 −Z1;2)), where X; Y; Z are the values of the landmarks’ coordinates. Every measurement was performed three times to compensate for the measurement error and the final value was calculated from the arithmetic mean (Arnqvist & Martensson, 1998). We calculated the percentages of the following parameters: traces (from the same specimen) included within parallel clusters and at least partly affecting the articular surface, trace category, shape, and location (anatomical area). The latter was divided into (i) low economy elements (less nutritive body parts) and (ii) high economy elements (more nutritive body parts), generally following the classification proposed by Drumheller et al. (2020). We also determined the percentages of the taxa represented by the studied specimens at the ‘clade’ (Macronaria, Diplodocoidea), ‘family’ (Diplodocidae, Camarasauridae, Dicraeosauridae) and ‘subfamily’ (only within Diplodocidae: Apatosaurinae, Diplodocinae) equivalent rank levels.

Figure 2 Examples of measurements of bite traces.

Examples of each possible measurement used for this article, taken on the traces A, B and C (visible in picture E from right to left) of pubis of AMNH FARB 675 (3D model of the holotype of “Apatosaurus” minimus, currently considered an indeterminate macronarian; ©American Museum of Natural History). (A) Length of trace B. (B) Width of trace B. (C) Depth of trace B, calculated from the cosine of the distance between the margins of the trace and its bottom. (D) Width of the entire bite area, calculated by measuring the distance between the external margins of the outermost traces A and C; in this case the length of the bite area is equal to that of the longest trace B. (E) Spacing between A and B, B and C.

To identify the potential bite makers on the sauropod bones, a survey of the dentition morphology of all carnivores from the Morrison Formation capable of leaving these tooth marks was done. Our dataset, which is mainly comprised of crown-based measurements published by Hendrickx, Mateus & Araújo (2015a), Hendrickx, Tschopp & Ezcurra (2020) and Hendrickx et al. (2020), includes information on crown length, thickness, elongation as well as denticle density along the mesial and distal carinae in a total of 293 in situ or isolated teeth belonging to four genus-level and six species-level (i.e., Allosaurus fragilis, A. jimmadseni, Ceratosaurus nasicornis, Marshosaurus bicentesimus, Torvosaurus tanneri and T. gurneyi) theropods from the Upper Jurassic of the United States and Portugal (see Data S4). The mesial dentition morphology, premaxillary tooth orientation, position of the carinae in both mesial and distal teeth, as well as the pattern of spalled surfaces resulting from tooth-to-bone contact and degree of crown wear along the dentition was also explored in each of these taxa. Crown-based measurements as well as the dental nomenclature and orientation follow the recommendations of Smith, Vann & Dodson (2005), Smith & Dodson (2003), and Hendrickx, Mateus & Araújo (2015b).

Results

In total, we identified 40 individual sauropods, collectively having 68 skeletal elements bearing bites, from literature and personal observations (Table 1). Among these, eight elements (pelvis from Camararaurus lewisi BYU 9047; femur from indeterminate diplodocine CMC VP7747; humerus, sacrum, right femur, right foot phalanx from Camarasaurus sp. GMNH-PV 101; left scapula from Galeamopus pabsti NMZ 1000011; left metacarpal from Galeamopus sp. WDC GB) could not be completely analyzed and, therefore, are omitted from the analyses below. The remaining 60 bones, belonging to 37 individuals, are split across four anatomical regions: spine (18 elements), chest/abdomen (15 elements), upper limbs (four elements), and lower limbs (24 elements), with one element (the apatosaurine caudal vertebra AMNH FARB 222-4) bearing bite traces on two different anatomical areas (both on the caudal neural arch and centra) and thus counted twice.

Table 1 Taxonomic identity of sauropod specimens bearing bite traces.

Number and percentages of tooth traces and sauropod individuals bearing them according to their taxonomy. At the clade-level, seven specimens, three classified as sauropod indet., one as neosauropod indet., and three as eusauropod indet., were referred as indeterminate (indet.). At the “Family”-level, all diplodocoid and macronarian specimens that could not be assigned to a “Family”-level clade are referred as indeterminate (indet.). At the “Subfamily”-level, only diplodocid specimens, either assigned to Diplodocinae, Apatosaurinae, or an indeterminate “Subfamily” level clade of Diplodocidae (indet.), are included. Note that within the analyzed specimens, diplodocids and diplodocoids are more represented than camarasaurids and macronarians, respetively. These may reflect various collection biases as well as potential genuine differences in population structure, but this should be considered when comparing the numbers of bite traces between various taxa here.

Clade	No Tooth traces	% Tooth traces	No Individuals	% Individuals	
Clade-level					
Diplodocoidea	194	55.7	19	51.4%	
Macronaria	82	23.6	11	29.7%	
indet.	72	20.7	7	18.9%	
Family-level					
Camarasauridae	65	18.7	9	24.3%	
Dicraeosauridae	62	17.8	1	2.7%	
Diplodocidae	96	27.6	13	35.1%	
indet.	125	35.9	14	37.8%	
Subfamily-level					
Apatosaurinae	75	78.1	6	46.2%	
Diplodocinae	20	20.8	6	46.2%	
Diplodocidae indet.	1	1	1	7.7%	

We found up to 48 tooth traces on a single bone and up to 12 traces within patches of parallel/sub-parallel traces (both on the ilium of the indeterminate eusauropod AMNH FARB 366), which were potentially produced by a single bite. Among the measured elements, the femur of the indeterminate apatosaurine AMNH FARB 222 bears both the largest bitten area (153 × 168 mm) and the largest single trace (128 × 28 mm, length × width), excluding removed traces. On the other hand, the smallest single trace (5.28 × 0.69 mm, length × width) was found on AMNH FARB 30066, a possible carpal bone of an indeterminate eusauropod.

As for tooth trace categories, we identified—among the 60 completely classified elements—a predominance of drags (174, 50.0%) and bite and drags (158, 45.4%) over pits (4, 1.1%), punctures (3, 0.9%) and removed traces (9, 2.6%). These bite traces were also classified according to the parameters listed below and the results are plotted as bar charts.

1. Shape (Fig. 3A): We divided the tooth traces in two categories based on their shape: straight and curved; this could give us information about the feeding style adopted by the carnivore.

2. Proximity to articular surfaces (Fig. 3B): The amount of traces located on articular surfaces or on other parts of the bone away from the ends.

3. Trace type (Fig. 3C): The category of the bite trace type.

4. Parallel clusters (Fig. 3D): The amount of traces included within parallel clusters which, according to D’Amore & Blumenschine (2009), are defined as groups of parallel traces close to each other and located on a similar area of the specimen.

5. Anatomical area (Fig. 4): We provide two groups of charts, the first one is made by counting the bite traces associated to each anatomical region, while the second one by counting the individual bones bearing such traces. The latter one was made considering that an abundance of tooth traces on the same element may just be the result of a single feeding event.

As noted, we analyzed 60 bones to produce these charts, however a fossil vertebra belonging to the apatosaurine AMNH FARB 222 was counted twice since it shows traces both on the neural arch and the centrum (which represents two different categories). For this reason, the total number of elements for the second group of charts is 61. Both groups of charts are divided between low and high economy elements, representing anatomical regions poor and rich in nutrients, respectively.

Based on these data, we inferred some information about the biology of the consumers and feeding style, along with the nature of the events that caused the tooth traces.

Figure 3 Charts of bite traces broken down by various categories.

Charts of the bite traces on sauropod bones, in each case divided by the number of analysed bones. (A) Chart showing the number and percentage of analysed bite traces divided by their shape (straight or curved) and category (drag, bite and drag, pit, puncture, removed). (B) Chart showing the number and percentages of bite traces found on articular ends of bones vs away from the articular ends. (C) Chart of bite traces by type. (D) Chart showing the number and percentages of bite traces seen in clusters vs isolated bones by bite type.

Figure 4 Charts to show distributions of bite traces by anatomical region.

Charts of the bite traces on sauropod bones, in each case divided by the number of analysed bones. (A) Chart showing the numbers and percentages of analysed bite traces found on low economy elements. (B) Chart showing the numbers and percentages of analysed bones bearing bite traces found on low economy elements. (C) Chart showing the numbers and percentages of analysed bite traces found on high economy elements. (D) Chart showing the numbers and percentages of analysed bones bearing bite traces found on high economy elements.

Discussion

Identification of the trace makers

The best way to identify the trace maker are teeth embedded in the bone or those that were likely lost during biting (e.g., Buffetaut & Suteethorn, 1989; Currie & Jacobsen, 1995; Jennings & Hasiotis, 2006; De Palma et al., 2013; Alonso et al., 2017). Unfortunately, no such embedded crown was found in any of the sauropod specimens under study here. Some authors such as Currie & Jacobsen (1995), Chure, Fiorillo & Jacobsen (1998), Rogers, Krause & Rogers (2003), and Happ (2008) either identified the trace maker based on the morphology and dimensions of its tooth marks or proposed the best candidate based on body size and/or abundance. Hone & Chure (2018), however, pointed out that many coeval predators with relatively similar ziphodont dentition would likely leave comparable traces on bones. Accordingly, ontogenetic and intraspecific variation could easily bias the analysis, and bites applied with different angles and worn/broken teeth would also affect the results (Hone & Chure, 2018), although careful consideration of spacing can allow for some inferences to be made with confidence (Brown, Tanke & Hone, 2021).

Another way to identify the trace maker was proposed by D’Amore & Blumenschine (2012) using the minute striations associated with a tooth mark and left by the denticles of the trace maker’s crown. Based on their observation of striated tooth marks made by the Komodo monitor Varanus komodoensis on the bones of carcasses, these authors provided a method to estimate the crown height, skull length, and body length of the bite maker with similar ziphodont dentitions like carnivorous theropods using the width of each striation (which relates to denticle size). Drumheller et al. (2020) applied D’Amore & Blumenschine (2012) method using the striations observed on bones belonging to Allosaurus, the ankylosaur Mymoorapelta, and indeterminate theropods and referred the bite marks showing the most widely spaced striations to large-bodied theropods from the Morrison Formation (see below). Our personal survey of the numerous bite marks left on sauropod bones revealed the scarcity of these minute striations, even in well-preserved bones with clear tooth marks. Only four tooth marks found in three sauropod specimens (i.e., an indeterminate eusauropod AMNH FARB 366 [ilium], and the Camarasaurus AMNH FARB 825 [caudal vertebra] and AMNH FARB 5760 D-x-128 [dorsal vertebra], Sc-3 [scapula], and Cd-o-21 [posterior caudal vertebra]) show striations reminiscent to those described by Drumheller et al. (2020) but we remain skeptical that they were made by the denticles of the crown’s maker. As already stated, such scarcity may be at least partially due to the way we performed the survey, however predatory archosaur-dominated assemblages are known to generally preserve few striations (Drumheller, D’Amore & Njau, 2023). In addition, Drumheller et al.’s (2020, table 1) published formulas to estimate the body length of the tooth makers using the striation’s width provided incoherent results, with estimated body length ranging from 108 to 671 m for striation widths of 0.7 to one mm (e.g., AMNH FABR 366, AMNH FARB 5760; see Data S5) using the formulas for distal denticles. We, however, noticed that Drumheller et al. (2020, table 1) formulas and results (Drumheller et al., 2020, table 5) do not match to estimate the body and skull length using the width of both mesial and distal denticles. Revising Drumheller et al.’s (2020) formulas to match their results still lead to incoherent results (46 to 290 m in body length using the formula for distal denticles) and their methods, therefore, appear to be inapplicable to estimate the body and skull lengths of the trace makers using striations made by coarse denticles (i.e., >0.6 mm or <8d/5 mm) while denticle width can reach up to one mm in particularly large ziphodont crowns such as those of Torvosaurus gurneyi and ‘Megalosaurus ingens’ (Malafaia et al., 2017a; Malafaia et al., 2017b; C.H. pers. obs., 2015). That being said, if the widest striations of one mm such as those seen in the tooth marks on AMNH FARB 366 and AMNH FARB 5760 were indeed made by denticles, Torvosaurus, the only theropod from the Morrison Formation with denticle width of up to one mm (see Data S5), can confidently be identified as the bite maker.

In all other non-striated bite marks (i.e., the large majority of our sample), and based on the width, depth and spacing of the tooth marks, it can at least be confidently suggested that those present on a number of specimens (AMNH FARB 92; AMNH FARB 222, 259, 264, 332, 392, 407, 582, 597, 642, 675, 5755, 5760, 5761, 6118, 30116, 30192; DINO 5119; TMP 1983.035.0003 (formerly UUVP 5309)) were left by large carnivores. The largest predators from the Morrison Formation are non-maniraptoriform avetheropods and the tooth marks were most likely made by these theropods. Chure, Fiorillo & Jacobsen (1998), Hone & Chure (2018), and Drumheller et al. (2020) also previously identified the bite marks found on sauropod specimens as belonging to large-bodied theropods. The extensive damage present on the indeterminate neosauropod AMNH FARB 264, on the two Camarasaurus specimens AMNH FARB 332 and AMNH FARB 582 (Fig. 5), on the indeterminate diplodocoid AMNH FARB 392, and on the indeterminate diplodocine AMNH FARB 642 further supports large theropods as credible candidates preying on diverse sauropod taxa.

Figure 5 Bite trace on a sauropod fibula.

A 3D model of the proximal end of a left fibula of Camarasaurus (AMNH FARB 582 ©American Museum of Natural History) in posteromedial view, showing a removed part of a bone, presumably from a particularly strong bite (A) and in close up (B).

The high percentage of parallel drags (Fig. 3D) suggests that the trace makers defleshed the carcass moving the head backwards, a pattern particularly common among large theropods based on their neck movements (Snively & Russell, 2007; Snively et al., 2013). Although teeth clearly came into contact with the bone surface, given the relative infrequency of tooth marks, such contact appears to be accidental (D’Amore & Blumenschine, 2009) rather than a systematic biting of the bone, as seen, by contrast, in tyrannosaurs (Hone & Rauhut, 2010). Although some theropods were capable of powerful bites (as suggested by the relatively high number of bite and drags and punctures), no evidence of bone gnawing (as defined by Capaldo & Blumenschine, 1994) was identified by us, and only six specimens (Camarasaurus AMNH FARB 332 and 582; Eusauropoda indet. AMNH FARB 366; Diplodocoidea indet. metapodial AMNH FARB 392 Apatosaurus AMNH FARB 550; Diplodocinae indet. AMNH FARB 642; Galeamopus WDC GB) bore extensive damage with parts of the bone removed.

Other terrestrial carnivores such as the small-bodied coelurosaurs Ornitholestes (Osborn, 1903), Stokesosaurus (Madsen, 1974), Tanycolagreus (Carpenter et al., 2005), Coelurus (Cope, 1887; Ostrom, 1980) and Hesperornithoides (Hartman et al., 2019) can be dismissed as possible candidates given that their crowns were not larger than 30 millimeters (C.H. pers. obs., 2023; Fig. 6) and could not have made such deep tooth marks on the sauropod bones because of limited jaw power. Likewise, the goniopholidid Amphicotylus, which is the largest crocodylomorph from the Morrison Formation, does not have ziphodont crowns capable of making these large striations (Drumheller et al., 2020). Instead, goniopholidids have a conidont dentition suited for impaling and holding prey items (Hendrickx et al., 2019), and these predators probably fed mainly on fish and small-bodied reptiles, dinosaurs, and mammals (Foster, 2020).

Figure 6 Morrison Formation theropods and their tooth crowns.

Theropod diversity and largest lateral crowns of theropods from the Kimmeridgian-Tithonian Morrison Formation of North America. (A) Crown of Allosaurus (UMNH VP 6105; CH = 50 mm) extrapolated to a CH (crown height) of 70 mm, the largest crown probably bore by the allosaurid Saurophaganax maximus (∼10 m in body length); (B) crown (BYUVP 725 12817; CH = 105 mm) of the megalosaurid Torvosaurus tanneri (∼9 m); (C) crown (CM 21703; CH = 58 mm) of the allosaurid Allosaurus sp. (∼7 m); (D) crown (ML 1828; CH = 75 mm) of the ceratosaurid Ceratosaurus sp. (∼6 m); (E) crown (DMNS 3718, Rmx4; CH = 41 mm) of the piatnitzkysaurid Marshosaurus bicentessimus (∼4.5 m); (F) crown (TPII 2000-09-29; CH 20 mm) of the tyrannosauroid Tanycolagreus topwilsoni (∼4 m); (G) crown (AMNH FARB 619; CH = 6.6 mm) of the maniraptoriform Ornitholestes hermanni (2 m); (H) crown (WDC-DML-001; CH = 6.4 mm) of the troodontid Hesperornithoides miessleri (0.9 m).

The large theropods from the Morrison Formation are currently represented by five unequivocal avetheropods, namely, Ceratosaurus nasicornis (Gilmore, 1920; Madsen & Welles, 2000), Marshosaurus bicentesimus (Madsen, 1976a), Torvosaurus tanneri (Galton & Jensen, 1979; Britt, 1991), and two species of Allosaurus, A. fragilis (Madsen, 1976b) and A. jimmadseni (Chure & Loewen, 2020). Saurophaganax maximus, considered by Smith (1998) to be a junior synonym of Allosaurus, also most likely represents a different allosaurid taxon from a higher stratigraphic level of the Morrison Formation (Chure, 1995; Foster, 2020). Referring tooth marks to any of these taxa is particularly challenging given the similarity of their dentition and the fact that subtly different actions of feeding can result in very different spacing of bite marks, making matches to tooth patterns in the jaws of these large theropods very uncertain (Hone & Chure, 2018). Hone & Chure (2018) for instance, tentatively ascribe the tooth marks found in the indeterminate diplodocoid DINO 5119 to Allosaurus sp. solely based on the much greater prevalence of this taxon on the fossil site. Likewise, Chure, Fiorillo & Jacobsen (1998) inferred that Torvosaurus or Ceratosaurus were the most likely candidates for making the tooth mark seen on a pubic foot of Allosaurus (AMNH FARB 813) based on the size of the bite and known tooth size in the largest theropods from the Morrison Formation. Drumheller et al. (2020) finally postulated that the tooth marks with closely spaced striations were made by Allosaurus and/or Ceratosaurus, and those with the largest striations by Torvosaurus, a very large size Allosaurus, and/or Saurophaganax based on the average denticle width measured on the crowns of these theropods and the formulas provided by D’Amore & Blumenschine (2012).

A thorough examination of the dentition of all large-bodied theropods from the Morrison Formation by one of us (C.H.) enables us to comment on these referrals and to provide additional information that may help identify the trace maker among these theropods. As correctly pointed out by Drumheller et al. (2020), difference in denticle size occurred between the four largest theropods here, which can help identify the trace maker using the striation spacing left by the denticles. The teeth of Ceratosaurus (UMNH VP 5278) and Allosaurus (UMNH VP 6499, 6239; CM 21703; AMNH FARB 851) with coarser denticles have a denticle density of no fewer than 8 to 9 denticles per five mm (here abbreviated in d/5 mm) in the mesial and lateral dentitions and for both distal and mesial carinae. Although Bakker & Bir (2004) noted that, among the three clades of large theropods from the Morrison Formation (i.e., ceratosaurids, allosaurids, megalosaurids), ceratosaurids have the smallest denticles (or the “finest serrations” sensu Bakker & Bir, 2004, p. 308) while those of allosaurids are intermediate in size, our dataset shows that the crowns of Ceratosaurus and Allosaurus have similar denticle densities (∼11 d/5 mm) on both mesial and distal carinae (Data S4). Conversely, as correctly observed by Bakker & Bir (2004), the largest crowns of Torvosaurus (BYUVP 725-12817; ML 1100; SHN.067; SHN.268) do have significantly larger denticles (5 to 6 mesial and distal d/5 mm in both the mesial and lateral dentition) whereas the lowest denticle density measured in Marshosaurus’ teeth (UMNH VP 6368; DMNS 3718) is 17–18 d/5 mm for the mesial carina and 14–15 for the distal carina (Data S4). Saurophaganax and the largest specimens of Allosaurus possibly had a slightly lower denticle density than that measured in our theropod tooth sample. However, it is unlikely that the crowns of the largest allosaurids had a denticle density lower than 7 d/5 mm as in Torvosaurus, a number comparable to the largest tyrannosaurids (e.g., Tyrannosaurus, Zhuchengtyrannus), which have the coarsest denticles in all theropods with ziphodont teeth (C.H. pers. obs., 2016). Based on this observation, striations spacing of more than 0.8 mm on the tooth mark were most likely made by the crowns of Torvosaurus whereas those between 0.6 and 0.8 mm, such as the largest striations measured by Drumheller et al. (2020) and tentatively referred to Torvosaurus or a particularly large allosaurid, could be made by Ceratosaurus, Torvosaurus, Allosaurus or Saurophaganax.

Another aspect that requires attention is the robustness of the crowns of these large-bodied theropods. The mesial and lateral dentitions of Marshosaurus are particularly labiolingually compressed (CBR < 0.6), which contrasts with the thicker mesial crowns of Ceratosaurus (CBR∼0.6–0.85), Torvosaurus (CBR∼0.65), and Allosaurus (CBR∼0.8–1.16). The mesial dentition of these three last theropods, therefore, appear to be less liable to break when contacting bone than those of Marshosaurus. The lateral crowns of Ceratosaurus are, however, strongly laterally compressed (CBR∼0.3–0.4) whereas those of Torvosaurus (CBR∼0.48) and Allosaurus (CBR∼0.65; Hendrickx et al., 2020) are thicker (Bakker & Bir, 2004; Hendrickx & Mateus, 2014; Data S4). The mesial and lateral teeth of Allosaurus are, in fact, particularly thick, to a point that this theropod is considered by Hendrickx et al. (2019) and Hendrickx et al. (2020) to have a pachydont dentition similar to that of derived tyrannosaurids. The latter are well-known to have incrassate and robust teeth adapted to bone-biting involving high mechanical stresses (e.g., Holtz, 2003; Snively, Henderson & Phillips, 2006; Reichel, 2010; Hendrickx et al., 2019). Both mesial and lateral dentitions of allosaurids would, therefore, appear to be better able to withstand tooth-to-bone contact than any other carnivorous theropods from the Morrison Formation, and the deepest and numerous tooth marks seen on the largest sauropod bones such as those seen in AMNH FARB 366 (a eusauropod ilium) were probably made by Allosaurus and/or Saurophaganax. Particularly long traces (>5 cm) may also be the result of an allosaurid feeding style which, according to Snively et al. (2013), employed their powerful neck muscles to rapidly move the head downward.

A survey of spalled surfaces and the degree of crown wear in the four large-bodied theropods from the Morrison Formation, reveals that all of the large theropod taxa from the Morrison Formation probably engaged in some tooth-to-bone contact during feeding. Tooth wear nevertheless indicates that the tip of the snout with the mesial dentition in Ceratosaurus, Marshosaurus and Torvosaurus were more often in contact with bones, whereas both the mesial and mesio-lateral/transitional dentitions of Allosaurus engaged in tooth-to-bone contact.

Crown apices of fully erupted premaxillary and mesial maxillary and dentary teeth are indeed usually worn out (i.e., the enamel has fully worn away to expose the underlying dentine) in Allosaurus fragilis and A. jimmadseni. The apices of distal maxillary and dentary teeth can also be slightly worn, but are more often intact in this genus, with the denticles often crossing the tip of the crown. Extremely worn out crowns with large spalled surfaces (differing from the wear facets due to tooth-to-tooth contact, which are common on the lingual surfaces of Allosaurus premaxillary teeth; C.H. pers. obs., 2016) have also been observed in some premaxillary (rpm1 and 3 of UMNH VP 1251), mesial maxillary (lmx2 of NHFO 455 and rmx2 of USNM 8335) and mesial dentary teeth (Ldt2 and Rdt3 of NHFO 455; Ldt3 of UMNH VP 6475) of Allosaurus but, this is not common. In the three other large Morrison theropods, a fully worn out apex of the crowns has been observed in the mesialmost dentary tooth of Ceratosaurus (Ldt1 of UMNH VP 5278) and one premaxillary tooth of Marshosaurus (Lpmx2 of DMNS 3718), indicating that the mesialmost teeth of these two taxa were frequently in contact with bones. Worn out apices are mainly found in the first premaxillary and dentary teeth in Ceratosaurus whereas the apices of the more distal crowns are intact or slightly worn out. This also seems to be the case in Marshosaurus based on the small sample available. Even when fully erupted, the lateral crowns of Torvosaurus either have the extremity of their apices worn out or intact apices (ML 1100, SHN.400). Fully worn-out apices are rare in the lateral dentition of Torvosaurus and have only been observed in one dentary tooth (i.e., ldt6 of BYUVP 725-12817). Two isolated shed teeth from Torvosaurus from the Late Jurassic of Portugal also have their apex worn out (SHN.215 and SHN 364; Malafaia et al., 2017a, fig. 11e). Based on their mesiodistal narrowness, elongation and/or thickness, these teeth most likely belong to the mesial dentition, suggesting that, unlike the lateral dentition, the mesial teeth of Torvosaurus were often in contact with bones.

Allosaurus also differs from the three basally-branching averostrans in its anteriormost snout morphology, which is mediolaterally broad (1.30–1.34 for the ratio of the lateromedial width divided by the anteroposterior length of the articulated premaxillae, maxillary processes excluded, in palatal view) and bears five mesiodistally wide premaxillary teeth (Fig. 7). The snouts of the megalosauroids Marshosaurus and Torvosaurus are comparatively narrower and anteroposteriorly longer (ratios of 0.795 and 0.760, respectively), whereas that of Ceratosaurus is anteroposteriorly shorter and mediolaterally wider (ratio of 0.89) but not as wide as that of Allosaurus (Fig. 7). Ceratosaurus, Marshosaurus, and Torvosaurus all bear three to four premaxillary crowns, which are particularly laterally compressed in the two megalosauroids (Fig. 7). A small space also separates the first premaxillary alveoli/crown (pmx1) from the left and right side of the cranium in Marshosaurus and Torvosaurus, whereas this space is wider in Ceratosaurus and Allosaurus, and noticeably wide in some specimens of Allosaurus. These dental variations in the four apex theropods from the Morrison Formation are reflected in the spacing of bite marks, especially when the theropod head moves parallel to the long axis of the skull, leaving even spaces between the marks (Fig. 7). We, however, agree with Hone & Chure (2018) that a head moving at a certain angle from its long axis during biting, as well as one or several misoriented crowns and unerupted, partially erupted, or missing teeth, directly affect the tooth mark pattern, making its identification particularly challenging (Fig. 7). Partly erupted teeth were in fact revealed to be common in the premaxillae of Ceratosaurus and Allosaurus (Data S4) and the fact that their apices are often perfectly intact shows that they did not participate in tooth-to-bone contact like the other fully erupted teeth. Therefore, only tooth marks showing a symmetrical pattern made of more than eight grooves with the two middle grooves being widely separated can be confidently assigned to allosaurids. Likewise, tooth marks with a symmetrical pattern of six grooves with the two middle grooves being particularly closely spaced are likely made by one of the two megalosauroids. Conversely, asymmetrical tooth marks showing deep furrows could be made by any of the apex theropods from the Morrison Formation.

Figure 7 Premaxilla and premaxillary dentition morphology and effect on the spacing between bite marks in the four apex predators from the Morrison Formation Ceratosaurus, Marshosaurus, Torvosaurus and Allosaurus (two specimens).

(A) Symmetrically duplicated premaxilla of Ceratosaurus (UMNH VP 5278), Marshosaurus (UMNH VP 7820), Torvosaurus (BYUVP 725 4882), and Allosaurus (Allosaurus specimen I: YPM PU 14554; Allosaurus specimen II: UMNH VP 20529) in ventral view; (B) In situ or reconstructed premaxillary dentition of Ceratosaurus, Marshosaurus, Torvosaurus and Allosaurus in ventral view showing the orientation of the mesial (in red) and distal (in blue) carinae on the crowns; (C–E) effect of tooth pattern and biting angle on the spacing between bite marks left by the premaxillary crowns of Ceratosaurus, Marshosaurus, Torvosaurus and Allosaurus when the head of the theropod moves (C) parallel to the long axis of the skull; (D) at an angle of 20 degrees from the long axis of the skull; and (E) at an angle of 45 degrees from the long axis of the skull.

Although Hone & Chure (2018) interpret the dentition of Allosaurus, Ceratosaurus and Torvosaurus as homodont, sharing similar gross morphology, the mesial dentition of Ceratosaurus, Allosaurus, and megalosauroids not only differ in the number of premaxillary teeth and the labiolingual thickness of the crowns, but also in the orientation of their carinae. The mesial carina is facing mesially and is strongly lingually displaced in the mesial dentition of Allosaurus (Hendrickx et al., 2020; Hendrickx, Tschopp & Ezcurra, 2020). On the other hand, the mesial carina is facing anteriorly and is centrally positioned on the mesial side of mesial teeth in Ceratosaurus, Marshosaurus, and Torvosaurus (and indeed all megalosauroids; Hendrickx, Mateus & Araújo, 2015a). Likewise, the distal carinae of Allosaurus and Ceratosaurus are strongly labially displaced and face distally in mesial teeth whereas those of megalosauroids are centrally positioned or weakly labially displaced in the mesial (and lateral) dentition and face linguodistally in the first two premaxillary teeth (Hendrickx, Mateus & Araújo, 2015a; Fig. 7). Consequently, the presence of striae on the lateral sides of multiple parallel tooth marks forming a symmetrical pattern likely results from the biting of an allosaurid. These striae would indeed be present in the tooth marks made by Ceratosaurus, Marshosaurus, and Torvosaurus if the snout of the latter moved at a strong angle from the long axis of the head, so that the anteriorly and posteriorly positioned mesial and distal denticles, respectively, would contact the bone during biting. Because mesial denticles are significantly smaller than the distal ones (DSDI > 1.2) in the mesial dentition of both Ceratosaurus and Marshosaurus (Hendrickx et al., 2019), a significant difference in the size in the striae from the two sides of a tooth mark would support these two taxa as the potential trace makers.

Chure, Fiorillo & Jacobsen (1998) discounted Allosaurus as a credible candidate of the large bite marks on a pubic foot of the same taxon (AMNH FARB 813) based on the fact that its crowns are significantly smaller than those of Ceratosaurus and Torvosaurus. With a crown height reaching up to 140 mm in some lateral teeth from Portugal (Malafaia et al., 2017b; Malafaia et al., 2017a), Torvosaurus bears among the largest crowns in all dinosaurs (Hendrickx et al., 2019; Fig. 6). The largest Ceratosaurus crowns we measured are 75 mm in height and likely did not exceed 100 mm, whereas those of Allosaurus are indeed shorter, with a crown height of 58.4 mm measured in the largest (and best preserved) crown (CM 21703; n.b., with a CH of 68.85 mm, rmx6 of SMA 0005/02 is the tallest crown measured for Allosaurus (Hendrickx, Mateus & Araújo, 2015a) but the specimen is particularly badly preserved so that measurements on the dentition of SMA 0005/02 should be seen as tentative; Fig. 6). As noted above, tooth wear suggests that the lateral teeth of Ceratosaurus and Torvosaurus probably rarely contacted bones whereas the mesial maxillary crowns of Allosaurus most likely did. Because the maxillary crowns from the largest allosaurid species probably exceeded 60 mm, we therefore consider Allosaurus and Saurophaganax as credible candidates of the tooth marks on AMNH FARB 813.

As for smaller traces, we did not observe any bisected or hook-shaped scores, which are typical for modern crocodiles (Njau & Blumenschine, 2006) and were likely produced by Morrison crocodyliforms, too (Hone & Chure, 2018; Hone & Chure, 2018). Similarly, the shape of the traces we recorded are different from what would be expected if they had been left by lizards (which usually produce curved traces, e.g., see D’Amore & Blumenschine, 2009), or mammals (e.g., West & Hasiotis, 2007; Longrich & Ryan, 2010). Early champsosaurs may be excluded too, considering they usually ate fish and that they possibly left traces similar to those of lizards and crocodyliforms (Foster, 2003; Foster, 2007; Hone & Chure, 2018). This suggests that the smaller traces were produced by small theropods, however we cannot tell if they were from small-sized taxa or juveniles of large theropods (since theropods were polyphyodont, Hendrickx, Mateus & Araújo, 2015b; Hone & Chure, 2018).

Collectively therefore, there is strong evidence that the various large bodied theropod taxa present in the Morrison Formation were feeding on sauropods. In at least some cases it is possible to rule in, or out, various taxa as candidates for given bite traces based on their size and shape, though this remains difficult in most instances because of the lack of details available or the multiple possible bite makers.

Palaeoecological implications

The wear seen in various large theropod teeth listed above may seem at odds with the relatively low numbers of bite marks seen on sauropod bones (compared to tyrannosaur faunas at least) and the idea that non-tyrannosaurid theropods largely avoided tooth-on-bone contact when feeding. However, this is not necessarily the case. When scavenging or feeding upon the carcass of a large sauropod, there would be literally tons of meat available for carnivores (see also Pahl & Ruedas, 2021) and it should be possible even for a number of large-bodied theropods to feed extensively on this without biting into bones so this may be less a case of active avoidance of such contact and more that it simply did not occur. As seen in Fig. 4, there are numerous bites on both individual elements and parts of the skeleton that we term high economy (potentially considerable muscle or meat attached) and low economy (little meat available). With the exception of the pelvic girdles however, there tend to be similar percentages of bites on both sets of areas, which is similar to the results found by Drumheller et al. (2020) (though their exact categories of high and low economy elements differs to ours). In terms of the numbers of elements that were bitten, a number of high economy elements show very few bites (pectoral girdle, fibulae) and some low economy elements are more often bitten.

Collectively, this is difficult to interpret as, for example, ribs might be readily damaged or destroyed in feeding and small elements like distal caudal vertebrae are rarely preserved so bites are not recorded even if present, and bones like the middle metatarsals might suffer few bites because they are covered by the lateral and medial ones. There are also far more low economy elements (e.g., there are five metatarsals for every fibula in the sauropod skeleton), and the high economy ones are often considerably larger. The pelvic girdles at least show a high number of bitten elements and a high number of individual traces seen. Given the attachment of so many large muscles to this region this appears to have been an area of attention from large theropods.

Although the exact nature of the bite distribution here is confounded by potential taphonomic and behavioural biases, they do suggest that bite marks are not especially rare compared to tyrannosaur faunas. This has been found at up to 14% of bones (Jacobsen, 1998) in tyrannosaur-dominated faunas and in our dataset here it is approximately 11% (68 bones out of over 600 viewed). The number here may well be inflated since one sauropod carcass may have multiple elements bearing traces, which is not a fair comparison to isolated elements, but this is still at least comparable to tyrannosaur faunas and does not represent a small fraction of the number of bites. The bites seen here are focused neither on major areas of meat (e.g., proximal limbs and girdles) or late-stage scavenging of poor quality areas like metapodials.

It is not clear how typical these values may be, however, given the ranges seen in some specific quarries within the Morrison Formation (see below). Among the systematically surveyed collection at AMNH, three localities are represented with the most material (Bone Cabin Quarry, Howe Quarry, Reed’s Quarry R), and numerous elements from the historic Cope collection may be from comparably productive localities in the Garden Park area (the exact provenance of the single bones excavated for Cope in this area is not always known; McIntosh, 1998) (see Tschopp et al., 2022). Among these localities, Bone Cabin Quarry clearly stands out in percentage of elements with bite traces, further emphasizing the impact of different taphonomies on such percentages. In short, it may be misleading to take this value of 11% as being an appropriate estimate for the Morrison or other sauropod-dominated faunas in general given the variation in exposure, taphonomy and collection (see below for more on possible ecological differences).

However, if large theropods were predominantly predating and feeding upon juvenile animals, as suggested by Hone & Rauhut (2010) among others, then tooth-on-bone contact would have been much more common. Here, theropods would potentially be breaking up and consuming most of the animal (which would partly explain their rarity in the fossil record) and would therefore involve the tooth-on-bone contact that could wear down tooth crowns. Young animals would have generally smaller and weaker bones that a large theropod could bite through and these would also often be incompletely ossified and with incompletely closed sutures that would make them easier to process (compared to an adult sauropod). In short, the relatively rare bites preserved on the bones of large sauropods were probably not causing the wear seen on large theropod teeth, these rather come from engagement with the destroyed (and so not preserved) bones of the more frequently consumed juvenile sauropods.

It is generally very difficult to determine whether tooth traces were the result of scavenging or predation events from bitten fossils (Holtz, 2003; Bader, Hasiotis & Martin, 2009; Hone & Chure, 2018). The latter may be clearly identified by the presence of healing tissue surrounding the trace, indicating a failed hunting attempt (Bell, Currie & Lee, 2012; De Palma et al., 2013) though perimortem injuries that differ from feeding traces could in theory be identified. A number of specimens (Camarasaurus AMNH FARB 332 and AMNH FARB 582; Eusauropoda indet. AMNH FARB 366 (Area 1 and 8); Diplodocoidea indet. AMNH FARB 392 (metapodial I); Apatosaurus AMNH FARB 550; Diplodocinae indet. AMNH FARB 642; Galeamopus WDC GB) were clearly damaged by powerful bites. However, there is no reason to think that these were the result of predation attempts.

It was suggested by Drumheller et al. (2020) (p. 13) that “bite marks on high economy bones are […] associated with predation, or at least early access to remains”, and we may attempt to identify which bite traces are potentially the result of predation events based on the anatomy of the predator and prey taxa. Predators typically aim at vital areas (like the hindquarters) to immobilize prey, so a manus or pes (as in Camarasaurus AMNH FARB 332 and the indeterminate diplodocoid AMNH FARB 392) would not represent an ideal target, being both hard and dangerous to try and bite on a fleeing or fighting animal. The necks of sauropods would not be especially high-economy areas as while there would be some musculature, it would be far less than regions such as the upper limbs and base of the tail, but a serious bite here could be damaging or fatal (Senter, 2007), if not actually easy to deliver on large animals at least (Taylor et al., 2011). The base of the tail would be a very high economy area for feeding and be an excellent point to strike to immobilize and even kill a sauropod targeting the caudofemoralis muscles and associated arteries, but such a density of muscle would be hard to penetrate with even multiple bites on a large animal and would not likely leave traces on the bones as a result of either a predation attempt or early stage carcass-consumption.

On the other hand, elements like the ilium of the indeterminate eusauropod AMNH FARB 366, the pubis of Galeamopus WDC GB or the dorsal vertebrae of Apatosaurus AMNH FARB 550 (Fig. 8), all of which were relatively large sauropod individuals, were probably out of reach even for a large theropod. Instead, the fibula (as in Camarasaurus AMNH FARB 582) and femur (as in the indeterminate diplodocine AMNH FARB 642) could represent optimal targets to seriously wound and, possibly, immobilize the prey by damaging the knee or ankle joints (Hunt et al., 1994). However, the trace on the femur is located on the medial condyle of the distal end, an area which would probably have been out of reach in a living animal. On the other hand, the fibula was damaged on the posterior corner of the proximal end, a more easily accessible location. If this was indeed the result of predation, the absence of healing tissue would suggest that the sauropod died shortly after being attacked (c.f., Carpenter, 1998; Happ, 2008). In short, there is no convincing evidence here of any bites being attributed to predation attempts on large sauropods rather than scavenging events.

Figure 8 Bite traces on a sauropod neural spine.

(A) Neural spine summit of the dorsal vertebra of Apatosaurus sp. (AMNH FARB 550) showing (B) extensive bite traces (in dorsolateral view). This region was unlikely to be the site of a predatory attack.

In the case of scavenging, more nutritious regions tend to be fed on first by carnivores or scavengers, such as the upper limbs, the chest and the abdomen and probably areas like the base of the tail. Not only do they represent attachment points for major muscle groups, but they may offer access to entrails too, also considering the location next to the anus/cloaca (Buffetaut & Suteethorn, 1989; Hunt et al., 1994; Jacobsen, 1998; Jennings & Hasiotis, 2006; Robinson, Jasinski & Sullivan, 2015). On the other hand, anatomical areas associated with low amounts of muscle, such as spinal elements and the lower limbs are generally the last to be consumed by carnivores (Hunt et al., 1994; Hone et al., 2010). In fact, less nutrient-bearing parts like the lower limbs may be preferably eaten only during times of low prey availability (Jacobsen, 1998). However, determining the difference between such late stage carcass consumption by a predator that made the kill, and a scavenger that has found a body may be impossible to determine without taphonomic evidence of transport, burial, or decay of the bones prior to the infliction of bites, though this is possible in some cases (e.g., Hone & Watabe, 2010).

Furthermore, there may be variation seen in patterns of carcass consumption and bone alteration based on seasonality and aspects of ecology such as prey density (in mammals at least, see e.g., Haynes, 1982) and lions are seen to deflesh carcasses to different degrees in open and closed habitats (Domıńguez-Rodrigo, 1999). Such aspects that may affect feeding traces are difficult, if not impossible, to account for in the dinosaurian fossil record and the variation seen in various Morrison Formation quarries may reflect these factors and others such as exposure time of carcasses. The Mygatt-Moore quarry of Colorado for example, shows high-levels of bite traces associated with scavenging with nearly 29% of the 2,000+ elements bearing theropod bites (Drumheller et al., 2020), while also having high numbers of traces from invertebrate feeders, (20% of elements, McHugh et al., 2020) that are not commonly seen, or at least recorded, in other Morrison quarries. In contrast, the Carnegie Quarry of Utah has over 2,000 known elements, but apparently only one recorded with bite traces (see Hone & Chure, 2018); similar numbers are known from the Howe Quarry (see Tschopp & Mateus, 2013; Tschopp, Mehling & Norell, 2020). The Cleveland-Lloyd Quarry of Utah shows only around 4% of elements bearing bite traces (Gates, 2005), though the quarry that has yielded over 10,000 elements is heavily over-represented by carnivorous theropods, with >60% being Allosaurus (Peterson et al., 2017) and so may not be representative of a normal system of activity. As such there is a wide variety of values known for bite traces and the inclusion of one or other of these localities with our dataset could inflate or reduce the relative number of traces across the entire Morrison Formation enormously.

Among our samples, a number of bites are in positions and on elements that would neither be possible during a predation attempt or likely inflicted during early stages of carcass consumption and so might represent scavenging. For example, in the indeterminate apatosaurine AMNH FARB 222, the preferred orientation of the traces on the tail vertebrae (oblique, posterodorsal to anteroventral) may indicate the employment of a specific defleshing technique (see Hunt et al., 1994) and, therefore, that they were the result of late stage consumption. This may also be true for the tooth traces on one rib of the indeterminate sauropod AMNH FARB 625 (aligned with the long axis) and the right scapula of Camarasaurus supremus AMNH FARB 5760 Sc-3 (showing a preferred orientation along the proximal margin, Fig. 9). The latter specimen also displays a significant number of traces in the same area, which would be nearly impossible to produce on a living prey.

Figure 9 Scapula of Camarasaurus supremus showing bite traces (A) A saruopod scapula (AMNH FARB 5760 sc-3) and close up (B) that shows bite traces that follow the preferred orientation along the distal margin of the bone.

Similarly, the traces on the caudal vertebrae of Camarasaurus supremus AMNH FARB 5760 (Cd–y–4) and C. grandis CM 11393 are located on their articular surface, an area accessible only after disarticulation, which can only really indicate late stage carcass consumption. Tooth traces located on the lower limbs (e.g., Diplodocoidea indet. AMNH FARB 597; Camarasaurus AMNH FARB 664 and also AMNH FARB 582, if not related to predation) and the spinal elements (e.g., Camarasaurus supremus AMNH FARB 5761) were also likely damaged during late stage feeding and may relate to scavenging, since they are associated with less nutritive areas. On the contrary, the bones from the upper limbs (e.g., Diplodocoidea indet. DINO 5119; Diplodocinae indet. AMNH FARB 660) and the chest/abdomen area (e.g., Macronaria indet. AMNH FARB 675) may have been bitten during early feeding, being located next to large muscles and entrails, although the general rarity of bites in these areas (Fig. 4) suggests that such feeding did not normally reach the bone.

We found a total of 120 tooth traces associated with articular surfaces (on a total of 348 traces, distributed among 60 elements), especially on femora (17), metacarpals (26) and metatarsals (54). They may have been the result of carnivores feeding on the cartilage caps surrounding such regions, but may also represent attempts at disarticulating the carcass as previously seen in the tyrannosaurines (see Hone & Watabe, 2010). The latter hypothesis could be true for those specimens bearing bite and drag traces (e.g., the indeterminate diplodocoid metacarpal AMNH FARB 92), punctures and removed parts. These types of traces are typically associated with stronger bites (D’Amore & Blumenschine, 2009; Hone & Rauhut, 2010), which could be expected in case of disarticulation.

Most of the studied fossils were attributed to diplodocoids and, more specifically, diplodocids (Table 1). At first glance, this does not seem to be controlled by a collection or sampling bias herein, because (1) camarasaurs are generally more abundant in the Morrison Formation (Foster, 2003); (2) because the diplodocoid-dominated bonebed from Howe Quarry (Brown, 1935; Tschopp, Mehling & Norell, 2020), which is mostly housed at AMNH and SMA (two collections we have studied extensively), has only produced a single bone with bite traces, and (3) because camarasaurs and diplodocoids are about equally represented in the AMNH collections, the collection we surveyed most systematically. But, although it would be easy to conclude that the prevalence of bite traces on diplodocoids over camarasaurs may indicate some sort of food preferences by Morrison theropods, we cannot exclude that our numbers are heavily impacted by a taphonomic bias. A more complete, systematic survey of Morrison localities with distinct taphonomic histories would be required to address taxonomic preferences, as is the case for the general abundance of bite traces across the formation.

Conclusions

Although the survey here was not exhaustive, it reveals that there are numerous bite marks present on large sauropod bones and that these are generally underrepresented in the scientific literature. There is potentially a rich source of data present here to help better understand the feeding habits and ecology of the large Late Jurassic theropods both in the Morrison and likely other formations as well.

Interpreting the bite traces on sauropod bones remains difficult. Despite the extensive survey here, the presence of multiple candidate theropod trace makers and the variation in time, space, deposit, and taxonomy of the sauropod faunas (as well as taphonomic and behavioural factors) means that drawing conclusions from this dataset as a whole is potentially problematic. However, it is clear that bite marks are more frequently produced on sauropods than previously realised and that there is wear to theropod teeth as the result of tooth-on-bone contact.

Ultimately this requires detailed study of individual specimens. The taphonomic history of the specimen and site is key to interpreting what the likely carnivore-consumed interaction was and information can be built up from there. Therefore, although this study points to the availability and importance of bite-traces on sauropods, the conclusions must remain tentative without considerable further study.

Supplemental Information

Supplemental Information 1 Table of specimens, bite traces, taxonomy and locality data

All the raw data of bite traces on sauropod bones, the taxonomic and anatomical identity of the elements and the basic calculations of abundances and references for all Supplemental Files.

Click here for additional data file.

Supplemental Information 2 Table of bite types and measurements

All the raw measurements of of the catalogues bite traces and their categorisation and documentation.

Click here for additional data file.

Supplemental Information 3 Table of measurements of AMNH FARB specimens

Specific catalogue of the systematic survey of the American Museum of Natural History collection of Morrison Sauropods

Click here for additional data file.

Supplemental Information 4 Crown based measurements of Morrison theropod teeth

All the measurements of theropod teeth used to determine the sizes of possible bites by theropods on sauropod bones and all of the raw data and the taxonomic identity of the specimens.

Click here for additional data file.

Supplemental Information 5 Body size calculations based on theropod teeth

Updated formulas and data from Drumheller et al. (2020) for body size estimation of trace maker.

Click here for additional data file.

Supplemental Information 6 MorphoSource 3D data

Click here for additional data file.

The authors thank Elisabete Malafaia for sharing photos of Torvosaurus teeth from Portugal. We thank Carl Mehling (AMNH) for assistance in collection access and specimen handling, Carolyn Merrill (AMNH) for training in and assistance with 3D scanning and model production, and Ayo Lewis for additional assistance in scanning.

Institutional Abbreviations

AMNH FARB American Museum of Natural History, Fossil Amphibians, Reptiles, and Birds Collections, New York, USA.

ANS Academy of Natural Sciences, Philadelphia, USA.

BYUVP Brigham Young University, Museum of Paleontology, Provo, USA.

CM Carnegie Museum of Natural History, Pittsburgh, USA.

CMC Cincinnati Museum Center, Cincinnati, USA.

DINO Dinosaur National Monument, Jensen, USA.

DMNS Denver Museum of Natural History, Denver, USA.

GMNH Gunma Museum of Natural History, Gunma, Japan.

ML Museu da Lourinha, Lourinha, Portugal.

MWC Museums of Western Colorado, Fruita, USA.

NHFO Natural History Fossil Collection, Qatar Museum Authority, Doha, Qatar

NMZ Natural History Museum Zurich, Switzerland.

SHN Sociedade de Historia Natural, Torres Vedras, Portugal.

SMA Sauriermuseum Aathal, Aathal, Switzerland.

TMP Royal Tyrrell Museum, Drumheller, Canada.

UMNH VP Natural History Museum of Utah, Salt Lake City, USA.

WDC Wyoming Dinosaur Center, Thermopolis, USA.

YPM Yale Peabody Museum, New Haven, USA.

Additional Information and Declarations

Competing Interests

Author Contributions

Data Availability

Emanuel Tschopp, Mathew J. Wedel, and David W.E. Hone are Academic Editors for PeerJ.

Roberto Lei conceived and designed the experiments, performed the experiments, analyzed the data, prepared figures and/or tables, authored or reviewed drafts of the article, and approved the final draft.

Emanuel Tschopp conceived and designed the experiments, performed the experiments, analyzed the data, prepared figures and/or tables, authored or reviewed drafts of the article, and approved the final draft.

Christophe Hendrickx conceived and designed the experiments, performed the experiments, analyzed the data, prepared figures and/or tables, authored or reviewed drafts of the article, and approved the final draft.

Mathew J. Wedel conceived and designed the experiments, authored or reviewed drafts of the article, and approved the final draft.

Mark Norell conceived and designed the experiments, performed the experiments, authored or reviewed drafts of the article, and approved the final draft.

David W.E. Hone conceived and designed the experiments, performed the experiments, authored or reviewed drafts of the article, and approved the final draft.

The following information was supplied regarding data availability:

The raw measurements of bite traces on bones and theropod teeth.

The 3D data are at available at MorphoSource:

- Metapodial [Mesh] [StrLight]: 10.17602/M2/M519465;

- Metatarsal [Mesh] [StrLight]: 10.17602/M2/M519458;

- Metapodial [Mesh] [StrLight]: 10.17602/M2/M519452;

- Phalanx [Mesh] [StrLight]: 10.17602/M2/M519445;

- Caudal Vertebra [Mesh] [StrLight]: 10.17602/M2/M519134;

- Astragalus [Mesh] [StrLight]: 10.17602/M2/M519128;

- Girdle Fragment [Mesh] [StrLight]: 10.17602/M2/M519032;

- Pubis [Mesh] [StrLight]: 10.17602/M2/M519030;

- Metacarpal V [Mesh] [StrLight]: 10.17602/M2/M519021;

- Metacarpal Iv [Mesh] [StrLight]: 10.17602/M2/M519006;

- Metacarpal Iii [Mesh] [StrLight]: 10.17602/M2/M519002;

- Metacarpal Ii [Mesh] [StrLight]: 10.17602/M2/M518998;

- Metacarpal I [Mesh] [StrLight]: 10.17602/M2/M518994;

- Scapula [Mesh] [StrLight]: 10.17602/M2/M518989;

- Femur [Mesh] [StrLight]: 10.17602/M2/M518983;

- Dorsal Rib [Mesh] [StrLight]: 10.17602/M2/M518977;

- Dorsal Rib [Mesh] [StrLight]: 10.17602/M2/M518973;

- Fibula [Mesh] [StrLight]: 10.17602/M2/M518963;

- Metatarsal I [Mesh] [StrLight]: 10.17602/M2/M518940;

- Metacarpal [Mesh] [StrLight]: 10.17602/M2/M518926;

- Metacarpal [Mesh] [StrLight]: 10.17602/M2/M518919;

- Phalanx [Mesh] [StrLight]: 10.17602/M2/M518787;

- Astragalus [Mesh] [StrLight]: 10.17602/M2/M518781.

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
