# Peer review of "Bite and tooth marks on sauropod dinosaurs from the Morrison Formation"

_PeerJ, doi:10.7717/peerj.16327_

## Round 0.1 · original submission · Major Revisions

The reviewers feel that your study is interesting. They, however, have various points for you to consider and suggestions for improvement.

Please, together with your unmarked revised manuscript, provide a marked-up copy as well as a document explaining how you have addressed every issue.

·

Basic reporting

I found this study intriguing and I applaud the authors for their work. My comments are relatively minor, and I make a suggestion that the authors expand their citations with a reference on taphonomy and paleoclimate overview within the Jurassic Morrison Formation (e.g. Dodson et al., 1980).

Experimental design

I think the authors could elaborate slightly on how they made certain measurements. I put a comment in the manuscript to that end.

Validity of the findings

The findings are interesting and worthy of publication. I would have liked to see the authors acknowledge some of the limits of their interpretation by drawing on analogs within the mammalian realm. For example, Haynes (1982) showed how tooth mark damage on mammalian prey can also vary based on seasonal variations in severity of the winters, abundance of prey, etc. (Haynes, G., 1982. Utilization and skeletal disturbances of North American prey carcasses. Arctic, pp.266-281.) I think a brief statement or two about the environmental unknowns that potentially underlie their samples would give this study a deeper grounding.

Additional comments

When I wrote my paper in 1991, it was with the hope that I would be able to develop a model similar to what these authors have done. I am very excited that the database has grown to the point that others, like these authors, can now propose such models.

Reviewer 2 ·

Basic reporting

Good, have made minor corrections in manuscript and suggested one additional reference.

Supplementary Excel file of specimens needs a column indicating at least locality name, member, state/county, and maybe lithology. I also suggest summarizing the full-sample range of these characteristics in the methods.

The Supplementary section absolutely needs a References list, as most of the citations in the Excel file of specimens do not overlap with the article text and thus are currently listed nowhere.

Only disappointment is that none of the bite traces could be even broadly attributed to any of the larger theropod genera or groups, although I appreciate the caution too.

Experimental design

All good.

Validity of the findings

All good.

Annotated reviews are not available for download in order to protect the identity of reviewers who chose to remain anonymous.

Reviewer 3 ·

Basic reporting

“Bite and tooth marks on sauropod dinosaurs from the Morrison Formation” by Lei and colleagues surveys these bone surface modifications in a comparatively understudied slice of geologic time. As such, this work represents a great addition to a growing body of literature examining trophic dynamics in terrestrial, Jurassic settings. The paper is largely well-written and organized, and the figures are well-made and communicate the major points of the paper well. The majority of my comments in the sections below center on expanding certain aspects of the methods and discussion, with an eye towards facilitating comparisons between existing and future surveys. This does include some new suggested citations, but I have incorporated them into my comments in the experimental design section.

Experimental design

Which specific Morrison sites were sampled for this study? I believe that the AMNH Morrison collection is largely composed of specimens from Bone Cabin Quarry, but I could be mistaken. This information is included for the tooth dataset, but I did not see it for the modified elements. The addition of another column to supplemental file 1 would be helpful with this regard. I particularly ask, because there seem to be interesting quarry-to-quarry differences in the Morrison, as presented in Drumheller et al., 2020, which is already cited, and a conference abstract we also presented on Bone Cabin Quarry material from the Museums of Western Colorado here:

J.B. McHugh, S.K. Drumheller, 2022. Assessing taphonomic activity among vertebrate remains from two Upper Jurassic Morrison Formation fossil sites: Bone Cabin and Mygatt-Moore Quarries, Society of Vertebrate Paleontology Annual Meeting book of abstracts, 59.

That’s obviously gray literature, but it seems like the results might end up aligning interestingly, which is good news for inter-analyst correspondence between two collections gathered from the same site.

Related to that, the authors point out that only two previous surveys of these bone modifications exist (Jacobsen, 1998 and Drumheller et al., 2020), but then in the paleoecological implications section of the Discussion, only the Jacobsen paper is compared to the new dataset. This seems like an odd oversight, especially with regards to the frequency of modified bones and scavenging vs. predation sections.

How does the AMNH sample alone, in which the authors systematically surveyed an entire, large collection, compare to the more incidental finds and literature mined specimens from other localities and museums? It might prove to be interesting to keep the existing, pooled dataset, but also compare frequencies and bone surface modification types between collections as well, for the same reasons stated above. I was also wondering why only sauropods were surveyed, given the prevalence of bite marks on theropod elements in other Morrison sites? How might including them change the resulting frequencies of bone surface modifications present in this collection?

D’Amore and Blumenschine’s (2012) method of estimating ziphodont trace maker body size from striation spacing in bite marks is mentioned at several points in the manuscript, but attributed to the paper in which it was employed to analyze Morrison specimens (Drumheller et al., 2020). The original paper should really be cited instead in those cases. I am also curious why the method itself was not employed in these samples. It could provide additional information to help support or eliminate potential actors, which as the authors discuss is particularly difficult to do in the Morrison. Denticle and striation/striae spacing is already discussed throughout, so it seems like plugging those measurements into the published formulae would require no additional primary data collection and minimal effort for some potentially informative results.

D'Amore, D. C., & Blumenschine, R. J. (2012). Using striated tooth marks on bone to predict body size in theropod dinosaurs: a model based on feeding observations of Varanus komodoensis, the Komodo monitor. Paleobiology, 38(1), 79-100.

Were all bite marks identified visually, or was magnification of any kind used in partnership with raking light? These methods can result in differences in bone surface modification reported frequencies, so adding a brief section with these details would be useful.

Blumenschine, R. J., Marean, C. W., & Capaldo, S. D. (1996). Blind tests of inter-analyst correspondence and accuracy in the identification of cut marks, percussion marks, and carnivore tooth marks on bone surfaces. Journal of archaeological science, 23(4), 493-507.

Are there concerns that the reliance on 3D surface scanning might also contribute to under-reporting smaller bite marks (both in terms of surface area affected, but also depth of tooth penetration)?

There are also a few additional publications the authors should consider including in their methods and conclusions, which would further bolster many of the patterns and conclusions presented in this study:

Support for interpretations of scavenging:

Jennings DS, Hasiotis ST. Taphonomic analysis of a dinosaur feeding site using geographic information systems (GIS), Morrison Formation, southern Bighorn Basin, Wyoming, USA. Palaios. 2006; 21: 480–492.

Longrich, N. R., Horner, J. R., Erickson, G. M., & Currie, P. J. (2010). Cannibalism in Tyrannosaurus rex. PLoS One, 5(10), e13419.

Support for interpretations of predation:

Xing, L., Bell, P. R., Currie, P. J., Shibata, M., Tseng, K., & Dong, Z. (2012). A sauropod rib with an embedded theropod tooth: direct evidence for feeding behaviour in the Jehol group, China. Lethaia, 45(4), 500-506.

Instead of rounding to 600 in line 117, what was the actual number of elements surveyed in this study? Percent of modified elements is a pretty common way to communicate degree of bone modification in an assemblage, and the authors do get into percentages based on the total number in the Discussion on line 474, where the total number is stated to be more like 800, which I assume includes all of the non-AMNH specimens as well.

In line 395 (and elsewhere, like figure 6), the authors discuss patterns of spacing between individual tooth marks in a serial bite as a means of differentiating potential actors. I greatly appreciate the detail here, but I was wondering if there were any concerns about repeated, overlapping serial bite marks complicating this method?

Brown, C. M., Tanke, D. H., & Hone, D. W. (2021). Rare evidence for ‘gnawing-like’ behavior in a small-bodied theropod dinosaur. PeerJ, 9, e11557.

Validity of the findings

Several of the questions I pose in the experimental design section above could have bearing on the findings drawn from the dataset, but I have no additional suggestions to add here.

Additional comments

Line 198: Are these serial marks, sensu Binford, 1981?

Line 218: Instead of scars, are these referring to striations, sensu D’Amore and Blumenschine, 2009?

Line 435 and after: Something has happened with opening and closing parentheses in this paragraph.

---

## Round 0.2 · Minor Revisions

Please, together with your unmarked revised manuscript, provide a marked-up copy as well as a document explaining how you have addressed the points raised by reviewer 3 in this second round.

Reviewer 3 ·

Basic reporting

The updates to this manuscript have added additional clarity and impact to an already strong study. What comments I do have are very minor.

Experimental design

To track down the source of the replicability issues the authors found in the striation-based calculations in Drumheller et al., 2020, and with the editor’s permission, I reached out to the coauthor who performed the calculations for that study. They confirmed the authors' suspicion that while the results presented were calculated correctly (i.e. using the formulae from D’Amore and Blumenschine, 2012), the formulae presented in the associated table were incorrectly printed. The authors have correctly identified the issue, specifically that there should have been a "+" instead of a "-" in front of the intercepts (b). The decreasing utility of the method with increasing striation spacing/body size is a separate problem, discussed in the original D’Amore and Blumenschine (2012) study as well. As the authors state, with the range of striation spacing they found, their calculated results ran into that issue.

Validity of the findings

The quote presented in line 599 “bite marks on high economy bones are […] associated with predation,” is taken from the following sentence: “Bite marks on high economy bones are therefore associated with predation, or at least early access to remains, while feeding traces on only low economy bones are interpreted to be caused by late access to remains, such as scavenging.” This method is more applicable to making an argument in favor of scavenging based on concentrations of marks on low economy bones, while marks on high economy bones suggest early access only and cannot differentiate predation from early access scavenging. It’s a quibble, but an important one, because these two behaviors remain very difficult to tell apart in the fossil record.

Additional comments

Is the hard return on line 186 intentional, or should the sentence starting on line 184 be folded into the following paragraph?

The authors use the word ‘scar’ a few times throughout the manuscript. That term has connotations relating to healing, and since nothing like that is currently noted amongst the marks they describe, they may want to use either ‘mark’ or ‘trace’ instead.

Line 256: Are the parallel clusters mentioned here equivalent to serial bite marks, sensu Binford, 1981?

While D’Amore and Blumenschine, 2012 was added to the text, it does not appear to have been added to the works cited section.

Do the copyright symbols in the AMNH images throughout the figure captions assign those rights to the museum because they figure AMNH specimens or because the images were taken by an AMNH employee? If it is the second option, that person should still be credited.

---

## Round 0.3 · accepted · Accept

Dear authors,

I have accepted your manuscript based on the reviewer recommendation (reviewer 3 from the previous rounds).

As you no doubt are aware, reviewer 3 is one of the authors of the original paper whose methodology you were trying to replicate. They have contacted PeerJ staff to let us know that they plan to publish an erratum on the detected error with the original journal. After discussion, the author has told us the erratum will not be submitted until after your paper is published (so that they can cite it as the source of the erratum).

Given that they found out about the issue while acting as a reviewer, reviewer 3 thought it best to be open with PeerJ about their intentions. And to let us inform you, so you are all 'kept in the loop'. We hope this ensures everything is transparent, and all parties know what is happening. If you have any questions, please do not hesitate to contact us.

Shortly you will be contacted by the production team to take you through the proofing stages.

Thank you again for choosing PeerJ as your publication venue. I hope you will use us again in the future.

Reviewer 3 ·

Basic reporting

All previously suggested edits either have been made or addressed by the authors.

Experimental design

No comments.

Validity of the findings

No comments.